# Direct chiroptical correlation of dissymmetric crystal morphologies

Qiang Wen[1], Melissa Tan[2], Ofir Eisenberg[1], Hadar Nasi[1], Vlad Brumfeld[3], Akash Tiwari[2], Hai-Mu Ye [2], Bart Kahr[2] ✉, Linda J. W. Shimon [3] ✉, Michal Lahav[1] ✉ & Milko E. van der Boom [1] ✉

The expression of molecular dissymmetry in crystal form has been studied since Pasteur correlated macroscopic and microscopic chirality by measuring the optical activity of solutions of enantiomorphous tartrate salt crystals. Here, we show a direct correlation between enantiomorphous metal-organic framework (MOF) crystals and the chirality of the molecular structure. The geometry of the habits is correlated with single-crystal optical activity along the accessible low-symmetry directions. Weak X-ray scattering from small crystals was consistent with a hexagonal, enantiomorphous space group. However, the heterochirality of the mirror image forms could not be established with X-rays, necessitating a different approach. The optical circular birefringence of the enantiomorphs as evidenced by chiroptical imaging with a complete polarimetric microscope, was used to correlate optical and morphological chirality.

The small but celebrated hemihedral crystal facets of sodium ammonium tartrate gave Pasteur, via manual resolution, evidence of dissymmetric molecular configuration only after solutions of dissolved crystals had been subject to polarimetry[1,2]. Differences in responses to left and right circularly polarized light are the dominant perturbations to the state of polarization in isotropic solutions of chiral compounds. While crystals of tartrate salts are transparent, their solid-state optical activity could not be measured straightaway in 1848. The optical activity of anisotropic crystals remains difficult to measure along low symmetry directions and requires accurate photometry and polarization modulation[3,4]. Only in 1997 were researchers able to establish the optical rotation (OR) anisotropy of enantiopure tartaric acid crystals[5], filling an apparent vacancy in the history of molecular chirality. These crystals, however, are not manifestly dissymmetric in form, like sodium ammonium tartrate. To the best of our knowledge, chiroptical measurements have never been made of sodium ammonium tartrate, in part because the crystals have remained difficult to grow and their enantiomorphism lacks the clarity of Pasteur's oft-reproduced drawings[6,7]. More often than not, chirality is veiled when dissymmetries are not manifest in the crystal habit. Therefore, the advent of a new system of enantiomorphous MOF crystals, with a clear

manifestation of dissymmetry, allows us to try to link crystal structure dissymmetry with habit. The break down of Friedel's intensity law can be used to determine the absolute chirality of crystal structures. However, anomalous scattering is unreliable for weakly X-ray scattering crystals that are small or of low quality. The assignment of absolute structure assignment by 3D ED (three-dimensional electron diffraction) is a rising remedy[8].

Here we report the crystal morphologies of a cadmium acetate complex of an organic ligand (AdDB)[9–16], hereafter compound QW-MOF. Enantiomorphous morphologies were formed of apparently twisted trapezohedra in which the top hexagonal pyramid appears to be rotated anti-clockwise (L) or clockwise (R) with respect to the bottom pyramid. The hexagonal axis is the vertical in the plane of the page (Fig. 1A). Over time, the bottom pyramid and top pyramid can become dissymmetric in size and shape (Fig. 1B). The best way to assign the morphology is {11$\bar{1}$} for isohedral faces on the bottom pyramid and enantiomorphous faces either {122} or {212}, on the top. Moreover, additional faces develop. We demonstrate the control of the morphological dissymmetry with growth conditions: the apparent off-set angle between the bottom and top pyramids can be varied by the solvent composition (Fig. 1C). The crystals are characterized by

[1]Department of Molecular Chemistry and Materials Science, Weizmann Institute of Science, Rehovot, Israel. [2]Department of Chemistry and Molecular Design Institute, New York University, New York, NY, USA. [3]Department of Chemical Research Support, Weizmann Institute of Science, Rehovot, Israel. ✉e-mail: bk66@nyu.edu; linda.shimon@weizmann.ac.il; michal.lahav@weizmann.ac.il; milko.vanderboom@weizmann.ac.il

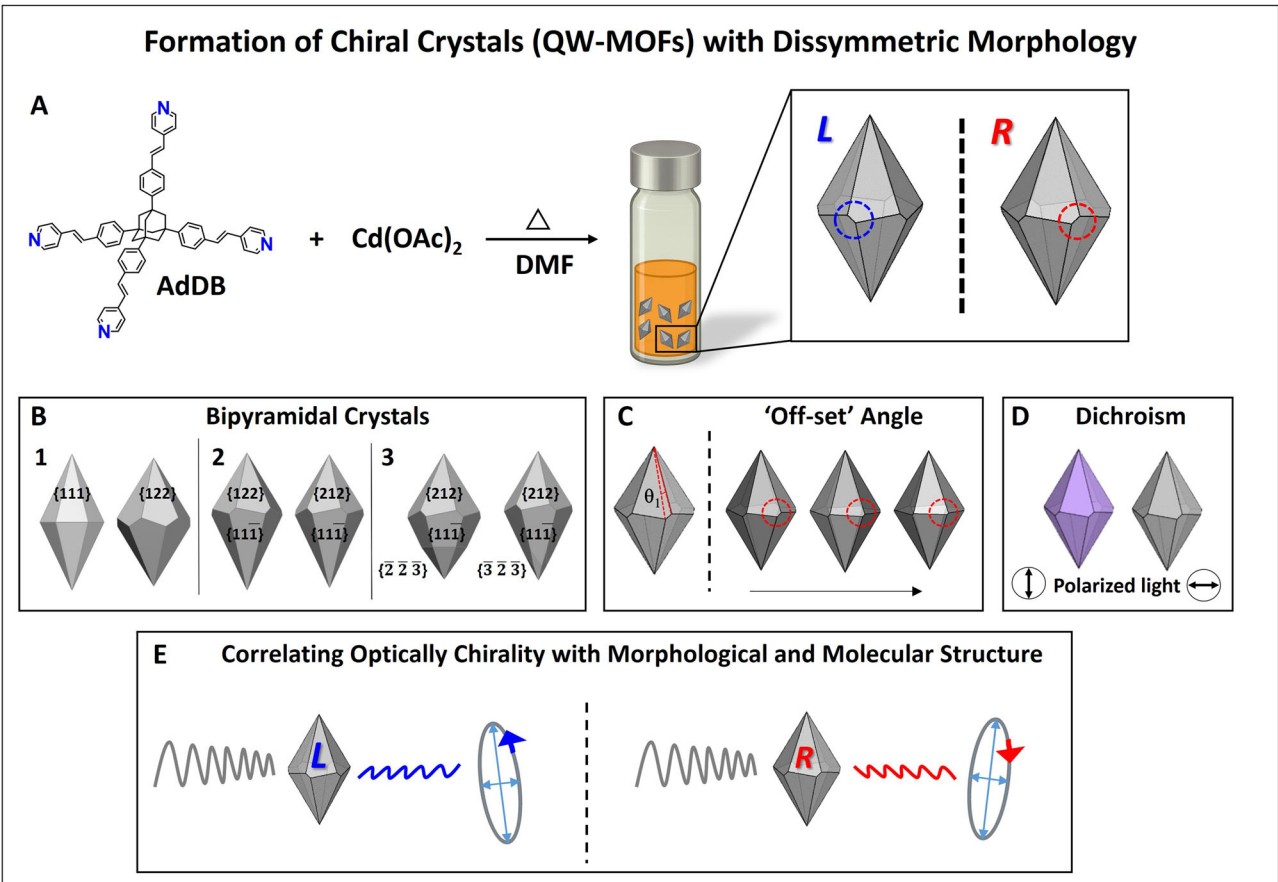

**Fig. 1 | Synthesis and crystal morphology. A** Reacting AdDB with a cadmium salt under solvothermal reaction conditions resulting in crystals with enantiomorphous morphologies of twisted trapezohedra. The crystals have helicoidal channels. **B** Bipyramidal crystals: (1) symmetry, (2,3) dissymmetry in size and shape between the top and bottom pyramids. 1-left: Drawing of isohedral habit with {111}. 1-right: Enantiomorph {122} crystal having sharp-zigzagging edges separating top and bottom pyramids. These two forms do not match experimental crystals. 2-left and right: Mirror images of dissymmetric polar bipyramids, where top $h \neq k$, and bottom $h = k$. 3-left: Isometric form {$\overline{2}\,\overline{2}\,3$} bisects lower pyramid. 3-right: additional face {$\overline{3}\,\overline{2}\,3$} observed in the bottom half of the pyramid resulting in curved pyramidal edges. **C** The apparent off-set angle between the bottom and top pyramids varied by the use of methanol. **D** The inclusion and alignment of sodium resorufin in the channels resulted in linear dichroism with the polarization of the dye absorption along the hexagonal axis. **E** Correlation of dissymmetric forms by equal and opposite measurements of circular retardance (CR) of single crystals.

extensive channels that can be functionalized with dye molecules (sodium resorufin). The alignment of the dye molecules resulted in strong linear dichroism under polarized light (Fig. 1D). The dissymmetric crystalline forms and crystalline optical activity could be correlated directly along the low symmetry directions presented by the as-grown enantiomorphous trapezohedra. We illustrate the correlation of dissymmetric forms by equal and opposite measurements of circular birefringence (CB) of single crystals in the solid-state (Fig. 1E).

## Results and discussion

Comparable crystals could be obtained by solvothermal synthesis in fused-quartz vials and by solvent diffusion (i.e., layering) of ligand and cadmium acetate solutions in quartz tubes at room temperature. The solvothermal growth was carried out in DMF at 105 °C for 24 h. We also layered two DMF solutions, one of ligand and one of cadmium acetate, with a solvent layer in between (DMF with $CH_3OH$ in some cases), in which crystals grew. These crystals were analyzed after 24 h. Optical microscopy, scanning electron microscopy (SEM), and X-ray microtomography (microCT) revealed the formation of the dissymmetric, chiral, trapezohedral crystals (Fig. 2). Ideal trapezohedra are staggered, achiral, bipyramids with 622 symmetry. Our enantiomorphs featured an apparent rotation or twist of the top pyramid with respect to the bottom (Figs. 2A–C, S1, S2, for example). The off-set angle of 5.8° (± 0.2) forms a right (R) or left (L) arrangement of the principal facets.

The size and shape uniformity of the crystals is high with typical dimensions of about 100 μm (L) × 60 μm (w). The handedness of 4600 crystals from five batches was established Pasteur like – by optical microscopy (49.4% R, 50.6% L; Table S1). The ensemble is essentially racemic. The growth of the crystals was monitored in situ by optical microscopy, at room temperature for more than 11 h (Fig. S3). The apparent bipyramidal off-set that defines the morphological dissymmetry was observable after 1 h when the crystals were still about one-fourth as long as their final size. According to symmetry rules, the faces {$hkl$} of enantiomorphous dihexagonal bipyramids must have $h \neq k$.

One of the best views of the morphology comes from X-ray microtomography (microCT; Fig. 2D). Some of these crystals have several curious features. (1) The bipyramids are longer on the bottom than on the top, as pictured in Figs. 1B2 and 2D. The length of the top pyramid is 75–80% of the bottom. (2) The bottom pyramid is more acute than the top, albeit this apparent angle depends upon the azimuth of the micrograph with respect to the vertical axis. The angle of the top pyramid is near 55° in projection, whereas the angle in the bottom pyramid is closer to 45°. These morphologies are consistent with the polar point group 6, as opposed to 622 where the top and bottom are equivalent. The sense of this axis could be a consequence of differences in mass transport that make some crystals appear polar. (3) The interfacial edges that separate the top and bottom pyramids

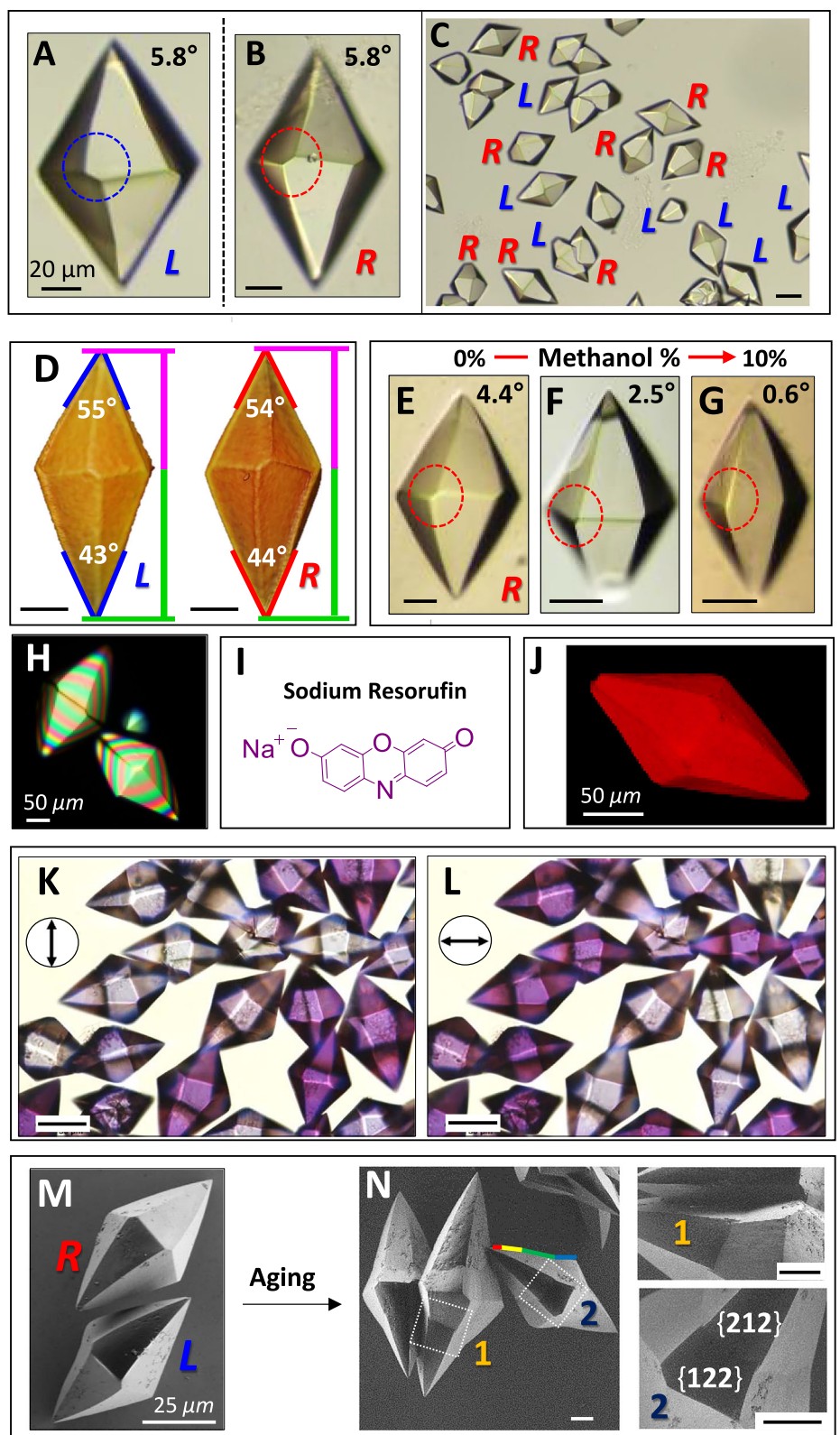

**Fig. 2 | Chiral hexagonal trapezohedral crystals.** Optical micrographs showing (**A**, **B**) Two crystals with opposite handedness, and **C** a racemic mixture, (scale bar: 50 μm). **D** Polar bipyramids: X-ray tomographs of enantiomorphs indicating that the top pyramid is shorter ($h \neq k$, pink vertical) compared with the bottom ($h = k$, green vertical) (scale bar: 20 μm). **E–G** Optical microscopy of enantiomorphous bipyramids prepared by layering using different amounts of methanol. The degree of the morphological dissymmetry increases with decreasing the amount of methanol (v/ v): 0%: 4.4° (±0.3), 5%: 2.5° (±0.5), 10%: 0.6° (±0.3) (scale bar: 25 μm). Optical images of the L-"twisted" crystals are shown in Fig. S6. **H** Images of crystals between crossed polarizers showing multiorder interference colors as a function of thickness. **I** Molecular structure of the dye. **J** Confocal fluorescence micrographs of crystals after adsorption with sodium resorufin ($\lambda_{exc}$ = 561 nm, $\lambda_{em}$ = 583–650 nm). **K**, **L** Linear dichroism of crystals under polarized light after adsorption with sodium resorufin, (scale bar: 50 μm). **M** Scanning electron micrographs (SEM) of crystals obtained after solvothermal reaction and N) after aging for 20 h at room temperature. These crystals show an additional form {122} producing a concavity. The bipyramidal edge curves, highlighted by a rainbow of line segments, widen the angle at the apex (scale bar: 20 μm).

have long and short bounding edges, horizontal and inclined, respectively. We have found that the best way to approximate this intersection[17] is with two types of pyramids: the bottom pyramid with isohedral faces $\{11\bar{1}\}$ and the top pyramid with enantiomorphous faces either $\{122\}$ or $\{212\}$. In contrast, symmetry 622 with twelve $\{111\}$ facets, i.e., symmetry 6 with six $\{111\}$ facets and six $\{11\bar{1}\}$ facets, would give a bipyramid with edges between the large faces of top and bottom pyramids nearly parallel to the intersecting planes. The crystal face assignments been validated by single-crystal X-ray diffraction (Fig. S4). Interpenetrated twinned crystals (RR, LL, and RL) were also observed as clearly seen by micro-CT (Fig. S5).

Strikingly similar chiral forms of centrosymmetric hexagonal ice were observed in the presence of anti-freeze proteins[18]. By analogy, we can deduce that enantiomorphous forms of the QW-MOF could likewise result from desymmetrization arising in the enantioselective adsorption of solution species, without the necessity of enantiomorphous crystal structures, and without facets assigned to heterochiral forms where $h \neq k$. The morphologies of QW-MOF are surely enantiomorphous but it is surprisingly difficult to be sure of the etiology of the morphological desymmetrization.

Chiral crystal shapes can form due to chiral molecules, crystal structures, and additives[19,20]. Ben-Moshe et al. found that screw dislocation-mediated growth of tellurium chiral nanocrystals leads to chiral polyhedral shapes[21]. The apparent off-set angle between the bottom and top pyramids in QW-MOF is affected by the nature of the solvent. Increasing the polarity of the reaction medium by the addition of 5–10% of methanol to the DMF resulted in an isohedral form. This observation shows that there is a family of closely related crystals forms, not all of which are evidently chiral (Figs. 1C, 2G, S6). The "offsets" of top and bottom pyramids with different amounts of methanol are shown in Fig. 2E–G: 0% methanol 4.4° (±0.3), 5% methanol 2.5° (±0.5), 10% methanol 0.6° (±0.3), as evidenced by the shrinking zone edge between the pyramid faces. The $^{113}$Cd NMR spectra of $Cd(OAc)_2$ indicated $CH_3OH$ solvation with a reduction in coordinated counter-anions, possibly altering $Cd^{2+}$ reactivity and crystal offset angle (Fig. S7). However, PXRD measurements confirm that the crystallographic structure remains unchanged (Fig. S8). Therefore, the achiral solvent composition only impacts the expression of the chiral crystal structure in the morphology. The optical anisotropy of the crystals with the optic axes lying at an angle to the optical path was indicated by the linear retardance (LR) between crossed polarizers (Fig. 2H). The crystals resemble quartz wedge compensators, displaying many orders of interference colors with increasing path length[22]. The crystals were extinct (dark) when the long axis (hexagonal optic axis) was parallel or perpendicular to the preferred plane of the polarizer and wave vector. Channels identified in the X-ray structure (*vide infra*) were established chemically. Immersing the crystals in a methanol solution containing a commonly used fluorophore, sodium resorufin (Fig. 2I), resulted in dye-inclusion crystals[23]. 3D fluorescence microscopy shows the homogeneous distribution of the red luminescence arising from the excitation of the purple resorufin fluorophore in the crystal (Fig. 2J). The resulting crystals are purple with a polarizer parallel to the long axes of the bipyramids and colorless in the orthogonal orientation (Figs. 1D, 2K, L). This complete linear dichroism is consistent with the resorufin ring system with a long-axis transition moment in the visible part of the spectrum aligned along the hexagonal c-axis of the MOF channels.

Over time, some crystals develop additional vicinal facets associated with reentrant angles (Fig. 2M→N). These facets on the top and bottom parts of the pyramids are best indexed as $\{122\}$, $\{12\bar{2}\}$, $\{121\}$, and $\{12\bar{1}\}$. They gradually curve inward at the apices. The continuous evolution of these facets is indicated by the curving edges, highlighted in Fig. 2N in a rainbow of tangent line segments consistent with the appearance of additional facets $\{3\bar{2}\bar{3}\}$ on only the bottom pyramid and results in curved pyramidal edges in some instances (Figs. 1B3, 2C, S2).

The AdDB ligand in other MOFs shows a tendency to form unusual shapes with decidedly curved edges; segmented tubes[9], yo-yo-like[12], flower-like[16], and brain-like[15] morphologies, consistent with adsorbate-induced deviations from ideal polyhedra. The gradual tapering of MOFs using this ligand and different nickel salts was well characterized previously[13].

Over 100 crystals from different batches were collected for single-crystal X-ray analysis at 100 K. The unit cell best fits a hexagonal cell: $a = b = 14.9799(2)$ Å, $c = 18.3087(3)$ Å. The resolution is poor; the scattering drops off quickly at $2\theta > ca.$ 23°. A MOF was constructed with the space group $P622$ using a crystal with a counterclockwise sense: a left (L) arrangement of the principal facets (Fig. 3). The Cd ions are coordinated by six pyridyl ligands and sit on sites of $D_3$ symmetry, corresponding to Wyckoff positions $d$ (1/3, 2/3, 1/2) and (2/3, 1/3, 1/2); the metal sits on a threefold axis[24]. The crystal has chiral (*left-handed*) hexagonal channels with a dimension of approximately 7.87 Å.

As the enantiomorph discrimination by anomalous dispersion of X-rays was equivocal, we turned to complete polarimetry. The differential scattering of circular polarized light is, in principle, sensitive to the crystallographic enantiomorphism. For most crystals, intrinsic optical activity or circular birefringence (CB = $n_L − n_R$) is given as the difference between the refractive indices for left ($n_L$) and right ($n_R$) circularly polarized light. This difference is typically much smaller than the linear birefringence (LB = $n_{//} − n_\perp$), the refractivity differences between orthogonally polarized linear vibrations. It is for this reason that Pasteur never tried to measure the optical activity of his transparent crystals along low-symmetry directions. Extracting the smaller quantities can be achieved by first determining the $4 \times 4$ polarization transfer or Mueller matrix (M) using an instrument that modulates the input and the output polarization[3,4,25–27]. M is reconstructed pixel by pixel from a light intensity signal captured with a camera, and the raw data of a complete imaging polarimeter, normalized to the magnitude of the first element $M_{11}$ as given in Fig. 4. The sixteen elements that construct M encode the linear optical anisotropies of the medium including the following: the linear extinction (LE), the linear retardance (LR), the circular extinction (CE), and the circular retardance (CR). For the transparent crystals, we are foremost interested in separating the dispersive quantities that introduce phase shifts, the retardances, and therefore we will not concern ourselves with extinctions that modulate amplitudes. Retardances are phase shifts that are linearly proportional to the intrinsic birefringences and the optical path length $z$: LR = $2\pi(n_{//} − n_\perp)z/\lambda$, and CR = $2\pi(n_L - n_R)z/\lambda$ where $\lambda$ is the monochromatic wavelength. Bipyramids were imaged in normal incidence transmission at 550 nm. The normalized Mueller matrices are displayed in Fig. 4. Matrices for enantiomorphous bipyramids are given in Fig. S9.

The polarization-dependent optical properties, |LB| and CB, can in principle be recovered through inversion of the Mueller matrix that reduces the convoluted optical properties to the differential properties[28,29]. This was achieved analytically[30]. However, ambiguities arise as the path length increases; the LR oscillates between $\pm\pi$, in principle, because whole-wave phase shifts of $n2\pi$, where $n$ is the integral order of the retardance, are not manifest. |LR| is encoded as an absolute distance that in practice oscillates between 0 and $\pi$, as a triangular waveform (Fig. 5A, B). Since the integral order is not known from polarimetry in a first analysis, we will maintain to the phenomenological language of retardance, |LR| and CR.

Comparison of the |LR| and CR profiles reveals an interdependence. As |LR| reaches $\pi$ radians, a rising edge is observed in the CR profile in dextrorotatory crystals (Fig. 5B). Similarly, for levorotatory crystals, maxima values of |LR| coincide with a falling edge in the CR function (Fig. 5A). Although linear anisotropy often obfuscates the recovery of chiroptical properties, in this case, it is beneficial, as the measurement captures several order changes beginning at an edge with an infinitesimal thickness[22]. Both the sign and magnitude of the OR (=CR/2) per millimeter can be extracted by interpreting the line

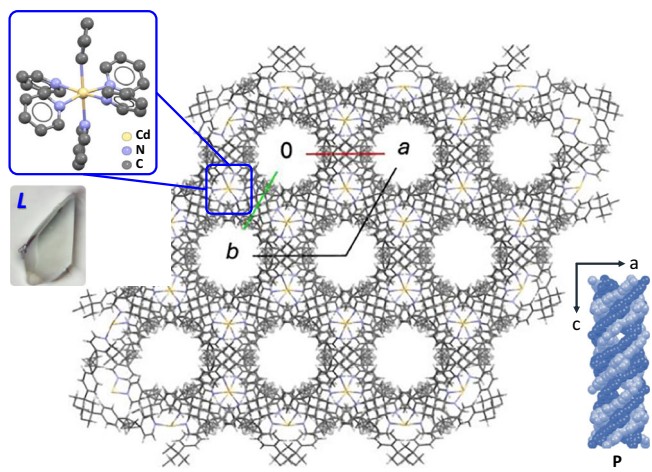

**Fig. 3 | Crystallographic model of QW-MOF in the space group *P*622 viewed along the *c* axis.** The photograph shows the crystal used for the shown data. On the right: a side view of an helical channel.

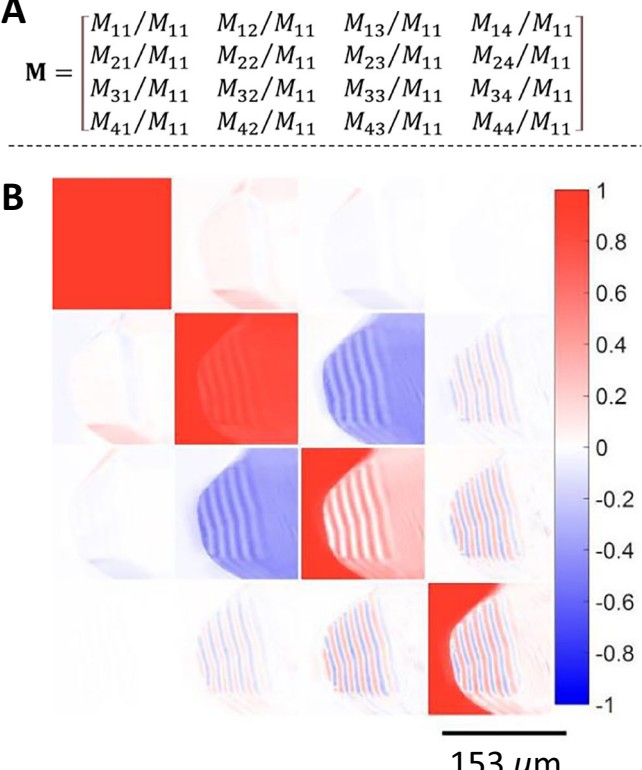

**Fig. 4 | Normalized transmission Mueller matrices. A**, **B** Representative example of a crystal with a right-handed morphological sense ($\lambda$ = 550 nm). The color scale represents real numbers varying between ±1. Each image describes pixel-by-pixel the transformation of an input polarization state to an output polarization state. Because the contributions to bianisotropic crystals from LR and CR are convolved in the raw matrix, further processing of M is required, thwarting a simple interpretation at this stage.

profiles corresponding to |LR| and CR = 2π(CB)$z/\lambda$. These patterns, over several orders, overcome a general limitation encountered in polarimetry, non-unique solutions to the differential Mueller matrix because of the periodicity of trigonometric functions (Fig. 5C).

If both the angle of the inclined plane ($\theta$) and the length of one period ($x$) are known, the magnitude and sign of the CR can be recovered. Taking the maximum repeating value of the CR as 0.09

radians, and extrapolating the height from zero to the first rise by the pyramidal angle (60°), the repeating linear distance between maxima is 12 $\mu$m according to Fig. 5D. The thickness, $z$, along any point can be determined by applying the function $\frac{x}{2} * \tan\theta = z$. In this way, the OR ca. 160°/mm is recovered, which is not an unusual value given the large spatial dispersion to be expected from a larger unit cell than that for other single crystals that have been analyzed along low symmetry directions[31,32].

Solving for the sign of OR is similarly straightforward. The oscillating sawtooth profile associated with CR displays a positive or negative slope that depends on the enantiomorph. In L-crystals the slope is always negative (resetting instantaneously at π) with increasing $z$. In *R*-crystals the slope is always positive. The chiroptical imaging technique provides spatially resolved information about optical activity across individual crystals. This technique allows us to distinguish between homochiral crystals and those potentially containing mixed enantiomorphs or domains arising from twinning[27]. Our imaging data reveal consistent oscillating circular sawtooth birefringence profiles across the entire crystal, which excludes racemic twins.

To establish that chiral bipyramids give qualitatively the same polarimetric properties as the QW-MOF, we examined first a chiral test crystal, ethylene diammonium sulfate (EDS), whose optical properties we have studied in detail[33–35]. EDS can have an isohedral tetragonal bipyramidal morphology but with perfect cleavage normal to the optic axis. The LB of EDS is smaller than that for QW-MOF, giving larger linear distances between orders in projection of 80 microns. The CR micrographs (See Figs. S10–S11), oblique to the optic axis, could be compared with those of QW-MOF qualitatively; the sense of the CR progression was dependent on the crystal chirality. The enantiomorph could be easily established in sections normal to the optic axis, where LR = 0 by symmetry. Moreover, the quantitative OR was found to be 15°/mm, which matches the value derived from the constitutive tensors for single crystals determined previously[33].

To conclude, while chiral crystal structures are numerous, the dissymmetry in structure is not always manifested in the morphology. Here, we analyzed an MOF whose dissymmetric crystal form can be tailored by the solvent composition. Pasteur famously had established a correlation between the macroscopic chirality of crystals evidenced by the disposition of hemihedral facets in dissymmetric crystals, and molecular configuration, evidenced by the action of the dissolved crystals on polarized light. However, he was unable to evaluate the optical activity of his transparent crystals directly due to the challenges associated with chiroptical measurements of crystals in general directions. We have correlated the optical activity with dissymmetric forms of the QW-MOF directly in low symmetry presentations by taking advantage of complete imaging polarimetry, a technology that had to await digital electrophotometry and electronic computers generations beyond Pasteur's toolkit.

## Methods

All reagents and solvents were purchased from Merck and used without further purification. 1,3,5,7-Tetrakis(4-((*E*)-2-(pyridin-4-yl)vinyl) phenyl)adamantane (AdDB) was prepared according to a literature procedure[36].

### Preparation of MOF Crystals

(a) Layering (solvent diffusion): A DMF solution (0.5 mL) of AdDB (3.0 mg, 0.0035 mmol) was added to an NMR tube ($\varnothing$ = 5 mm, $l$ = 178 mm). Next, DMF (0.25 mL) and 0.5 mL solutions of Cd(OAc)$_2$·2H$_2$O (2.8 mg, 0.010 mmol) in (a) DMF, (b) DMF:CH$_3$OH = 19:1 (v/v), or (c) DMF:CH$_3$OH = 9:1 (v/v) were added. The tube was sealed and left at room temperature for 24 h. The degree of the twist angle decreases with increasing the amount of methanol: 0%: 4.4° (±0.3), (b) 5%: 2.5° (±0.5), (c) 10%: 0.6° (±0.3).

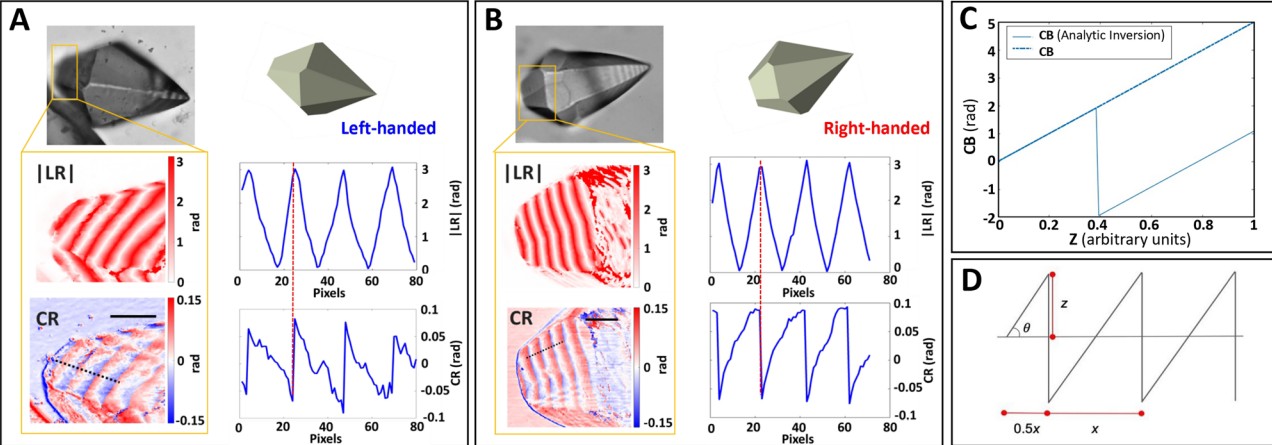

**Fig. 5 | Interdependence of bianisotrop.** The off-optical axis of (**A**) L-crystals and (**B**) R-crystals were analyzed as indicated by the areas enclosed in orange rectangles. Both linear retardance (LR) and circular retardance (CR) images ($\lambda = 550$ nm) are shown. Scale bar: 50 μm. The 1D plots are drawn along the path shown in the |LR| and CR images indicated with a black dash line. The vertical red dash-lines at the maximum of the |LR| indicate the changes from maximum to minimum in the CR. **C** Comparison of intrinsic CB (dashed line growing linearly thickness) versus observable quantity calculated through analytic inversion (solid line) which oscillates between 0 and π radians. **D** Simulation of CR profile. The optical path length $z = \frac{x}{2} * \tan\theta$.

(b) Solvothermal conditions: A DMF solution (1.0 mL) of AdDB (6.0 mg, 0.0070 mmol) was added to a 20 mL fused-quartz vial. Next, a DMF solution (0.5 mL) of Cd(OAc)$_2 \cdot$2H$_2$O (2.8 mg, 0.010 mmol) was added. The vial was heated in an oven at 105 °C for 24 h. A transparent solution was obtained after cooling down (the temperature of the oven was reduced by 10 °C every hour) to 25 °C. The solution was kept at 25 °C for 24 h, resulting in the formation of crystals on the surface of the vial which were collected by using filtration under reduced pressure. A white powder was obtained after drying under vacuum (~$10^{-2}$ mbar, 82% yield based on AdDB). Larger crystals can be obtained by decreasing the concentration of the starting materials by adding 0.5 mL of DMF (Fig. S2).

### Dye inclusion
The colorless crystals (5.0 mg) were immersed in a methanol solution (4.0 mL) containing sodium resorufin (1.2 mg) for 15 min. The purple colored crystals were washed with methanol until there was no detectable fluorescence from the washing solutions.

### Single crystal X-ray diffraction
A single crystal was coated in Paratone oil (Hampton Research) and mounted on MiTeGen loops. It was flash frozen in a liquid nitrogen stream of the Oxford Cryostream. Diffraction data were measured at a low temperature of 100(2) K using Cu K$\alpha$ $\lambda = 1.54184$ Å on a Rigaku Synergy R diffractometer equipped with a HyPix Arc-150° detector. The Rigaku data were processed and reduced with CrysAlisPro 1.171.41.111a[37]. The structure was solved with SHELXT-2018/2[38] and refined with SHELXL-2016/4[39]. All non-hydrogen atoms were refined anisotropically, and hydrogens were placed in calculated positions and refined in riding mode. The solvent masking protocol of Olex2[40] was run. The crystal data and the structural refinement are summarized in Table S2. The hexagonal channels contain DMF and acetate counterions and were not resolved. Lowering the symmetry, consistent with the morphological polar symmetry 6, was no remedy for the refinement.

### Optical microscopy
Optical micrographs were taken using a Nikon E600 Pol microscope and a Nikon DS Fi1 camera, or with Olympus BX50 and BX53 microscopes equipped with Olympus DP74 and SC50 cameras.

### Mueller matrix microscopy
Crystals for polarimetry were mounted between glass in a 1:1 mixture of immersion oil (RI = 1.7) and DMF to minimize scattering at interfaces. The microscope design is described elsewhere[41–43], based on previous efforts[44]. In short, continuous rotation of a pair of quarter-wave plates, with respect to stationary linear polarizers, generates a smoothly varying, time-dependent intensity signal that is demodulated to recover M, a sixteen-element transformation matrix that relates the polarization state of an incident beam of light, expressed as a four-element Stokes vector $S$, to its corresponding output vector: $S_{out} = MS_{in}$. Data are displayed as a 4 × 4 array of images representing the spatial variance of one of the elements $M_{ij}$ of the matrix (normalized by $M_{11}$). The microscope was built from a Zeiss ZM1 inverted microscope to which the home made mechanical modulators were added in the transmitted light path before and after the sample. Intensity measurements were collected with a Hamamatsu Flash Orca camera. The elements of M are real-valued, measurable, and completely describe the polarization-dependent linear optical effects. In general, each element of M represents a convolution of contributions from several optical properties; consequently, linear and circular anisotropies cannot be interpreted from M directly but must be recovered through matrix decomposition methods[30].

### Confocal fluorescence microscopy
Confocal imaging was conducted by using an upright Leica TCS SP8, equipped with internal Hybrid (HyD) detectors and Acusto Optical Tunable Filter (Leica microsystems CMS GmbH, Germany). Dye excitation was performed using a 561 nm laser at 1.0% power. Emission signal was collected using an internal HyD detector at the range of 583–650 nm. Images were acquired using the galvometric scanner with a 20X air objective (HC PL APO 20X/0.75 CS2), providing images with FOV = 123.1 mm, pixel size = 0.152 mm. Z stacks were acquired using the galvo stage, with 0.281 μm intervals. The acquired images were visualized using ImageJ.

### Scanning electron microscopy (SEM)
SEM measurements were performed using HRSEM ULTRA-55 ZEISS VP ZEISS instruments at an EHT voltage of 1.5 kV. SEM samples were prepared by placing a drop of the crystals dispersed in ethanol on a silicon substrate, which was dried under air.

## X-ray microtomography

The data were obtained with X-ray micro computed tomography instrument ZEISS Xradia 520 Versa. X-ray source voltage was set to 40 kV. Transmitted X-rays were converted into visible light with a scintillator and magnified with a 40× objective lens. The recorded pixel size was 0.33105 $\mu m$. The detector to sample distance was 22 mm. The X-ray source to sample distance was 20 mm. For each sample 801 X-ray transmission images were collected at the rotation axis angle spanning 360 degrees. These transmission image series were reconstructed into a three-dimensional tomographic dataset using the control software supplied with the instrument.

## Data availability

The data that support the findings of this study are available within the Article and its Supplementary Information files. Crystallographic data for the structure reported in this Article has been deposited at the Cambridge Crystallographic Data Center, under deposition number CCDC 2368214. Copies of the data can be obtained free of charge via https://www.ccdc.cam.ac.uk/structures/.

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

## Acknowledgements

This work at WIS was supported by the Horizon 2020 Research and Innovation program of the European Union under the Marie Sklodowska-Curie grant agreement no. 642192. This research was also supported by the Tom and Mary Beck Center for Renewable Energy as part of the Institute for Environmental Sustainability (IES) at the Weizmann Institute of Science. H.N. is an IES Fellow. M.E.vd.B. holds the Bruce A. Pearlman Professional Chair in Synthetic Organic Chemistry. The research was funded by Minerva and the Israel Science Foundation. We would like to thank Prof. Leslie Leiserowitz and Prof. Alexander Shtukenberg for fruitful discussions. Work at NYU was supported by the US National Science Foundation (ENG-2325911, DMR-2003968, DMR-1420073) and by a Sokol Fellowship from the NYU Department of Chemistry to M. Tan.

## Author contributions

Q.W., M.L., B.K., and M.E.v.d.B. designed the experiments. Q.W., O.E., H.N., V.B., L.J.W.S., M.T., A.T., and H-M.Y. performed the experiments and analyzed the data. Q.W., M.L., L.J.W.S., B.K., and M.E.v.d.B. prepared this manuscript.

## Competing interests

The authors declare no competing interests.
