## [Transparent Peer Review file · Nature Communications]

Direct Chiroptical Correlation of Dissymmetric Crystal Morphologies

Corresponding Author: Professor Milko van der Boom

Version 0:

Reviewer comments:

Reviewer #1

(Remarks to the Author)

Establishing a characterization method that effectively links the morphological chirality of materials with molecular chirality or topological chirality is of significant importance for understanding the growth mechanisms of chiral materials across both microscopic and macroscopic scales and for controlling their structure and chirality. This manuscript reports an example of QW-MOF with enantiomorphous morphologies being synthesized by controlling the synthesis conditions. The crystals crystallize in a chiral space group, and the optical activity is directly correlated with the QW-MOF's dissymmetric forms using Mueller matrix polarimetry. The material's performance is impressive, but the study on the structure-activity relationship is not quite detailed enough. Some issues still need to be clarified before the final determination for acceptance. However, before the manuscript can be accepted for publication, further clarification is needed on the reasons behind the formation of morphological chirality and whether the optical activity originates from the crystal structure's absolute chirality or the dissymmetric morphology itself. The specific issues to be addressed are as follows:

1. On page 2, line 54, the statement "The best way to assign the morphology is {111} for isohedral faces on the top pyramid and enantiomorphous faces either {122} or {212}, on the bottom" is incorrect; as the top pyramid and bottom pyramid are reversed. Additionally, has the reliability of the crystal face assignments been validated by single-crystal X-ray diffraction characterization?
2. In the introduction, the authors suggest that enantiomorphous MOF crystals can help us understand the link between symmetry and crystal habit. However, MOFs with such enantiomorphous morphologies are rare, as demonstrated in this work, while a few examples of inorganic crystals exhibit similar features (Science 2021, 372, 729–733). Do MOF crystals offer additional advantages, such as size, unique optical properties, or ease of morphology tuning?
3. In previous reports, the authors synthesized MOFs formed by 3d metal ions and the non-chiral AdDB ligand with different morphologies (Angew. Chem. Int. Ed. 2022, 61, e202205238). Mn-AdDB and Cd-AdDB (QW-MOF) crystallize in the chiral P622 space group, but Mn-AdDB exhibits symmetric hexagonal morphology. A discussion on the differences in coordination geometries among various metal centers and the role of metal ions in forming enantiomorphous morphologies is necessary.
4. In many examples of spontaneous chiral resolution, the Flack parameter is typically used to differentiate between enantiomorphic crystals. The Flack parameter of QW-MOF is 0.28, which could be due to small crystal size or poor diffraction quality, but it is also possible that some racemic twinning is present. Could chiroptical imaging with a complete polarimetric microscope be used to determine the purity of the crystal's chirality? Additionally, can the twisted bipyramidal crystals possibly arise from racemic twinning?
5. Authors demonstrate the correlation between methanol proportion and crystal offset angle, and mention that solvent polarity might be one of the reasons. However, it lacks further experimental and theoretical support.
6. The oscillating sawtooth profile associated with CR displays a positive slope for R-twisted crystals or a negative slope for L-twisted crystals. What about the results for symmetric bipyrimidine crystals or crystals with an offset angle close to 0? Is the difference between L- and R-twisted crystals attributed to the morphological chirality, or does it originate from the topological chirality (the P or M helical structure in L- or R-twisted crystals)?

Other:

1. On page 2, line 36, "is remains" should be "remains".
2. On page 3, line 73, "3-left" should be corrected to 3-right.
2. In Table S2, the molecular formula should also include acetate counterions, and the decimal points for the R1 and wR2 values need to be corrected.

Reviewer #2

(Remarks to the Author)

In this work, the authors demonstrate a direct correlation between enantiomorphous MOF crystals (named QW-MOF) and chirality at the molecular level. The crystal habits' geometry is linked to single crystal optical activity along low symmetry directions. The X-ray scattering from small QW-MOF crystals have been confirmed a hexagonal, enantiomorphous space group. In addition, the dissymmetric crystal forms of QW-MOF structures can be customized by solvent composition. Interestingly, the optical circular birefringence of the MOF enantiomorphs, observed through chiroptical imaging with a polarimetric microscope, has been used to correlate optical and morphological chirality. These results provide a new way to understand the molecular chirality of MOFs with their dissymmetric morphology. The reviewer encourages the acceptance of this manuscript after minor revisions.

1. The authors' method can only differentiate the correlation between crystals with distinct symmetrical morphologies. How does this differ from using polarizing microscopy to distinguish enantiomeric crystals?
2. The molecular-level helical chirality of QW-MOF materials needs confirmation through X-ray single crystal diffraction. How does this directly relate to its macroscopic morphology?
3. The MOF crystals in this study are highly perfect, and their morphology's dissymmetry correlates well with molecular chirality. Are the techniques and methods developed in this work also applicable to systems where crystals do not show obvious dissymmetry in their morphology?
4. In Table S2, the value of 'Final R [$I > 2(I)$]' and 'R (all data)' are written incorrectly., e.g., 'R1 = 0. 0.2950, wR2 = 0. 0.5856', 'R1= 0.0.3109'.

Version 1:

Reviewer comments:

Reviewer #1

(Remarks to the Author)

The author has answered my questions appropriately, so the manuscript is now acceptable for publication.

Reviewer #2

(Remarks to the Author)

The authors have revised the paper quite well, and I recommend its acceptance in its present version.

RESPONSE TO REVIEWERS' COMMENTS

Reviewer #1 (Remarks to the Author):

Establishing a characterization method that effectively links the morphological chirality of materials with molecular chirality or topological chirality is of significant importance for understanding the growth mechanisms of chiral materials across both microscopic and macroscopic scales and for controlling their structure and chirality. This manuscript reports an example of QW-MOF with enantiomorphous morphologies being synthesized by controlling the synthesis conditions. The crystals crystallize in a chiral space group, and the optical activity is directly correlated with the QW-MOF's dissymmetric forms using Mueller matrix polarimetry. The material's performance is impressive, but the study on the structure-activity relationship is not quite detailed enough. Some issues still need to be clarified before the final determination for acceptance. However, before the manuscript can be accepted for publication, further clarification is needed on the reasons behind the formation of morphological chirality and whether the optical activity originates from the crystal structure's absolute chirality or the dissymmetric morphology itself. The specific issues to be addressed are as follows:

Answer: We appreciate the insightful feedback on our manuscript. Below, we address the key issues raised regarding the relationship between morphological chirality, absolute chirality, and optical activity in our QW-MOF.

1. On page 2, line 54, the statement "The best way to assign the morphology is {111} for isohedral faces on the top pyramid and enantiomorphous faces either {122} or {212}, on the bottom" is incorrect; as the top pyramid and bottom pyramid are reversed. Additionally, has the reliability of the crystal face assignments been validated by single-crystal X-ray diffraction characterization?

Answer: The referee is correct and his/her attentiveness is much appreciated. The face families should be {111} and {12-2} and this has been corrected. We have included in the Supplementary information a photo of an

X-ray measured crystal, mounted on a loop, with the crystal axes delineated and showing that the faces can be indexed with those families (Page S5, **Fig. S4**)

Fig. S4. Example of morphology planes analysis (hkl) of **QW-MOF** using solvothermal conditions. The optical microscopy image show a crystal mounted on a loop. The individual planes $(1, 0, 1)$ and $(2, \bar{1}, \bar{2})$ were indexed by CrysAlisPro. The crystallographic axes are marked in blue. Note that since the symmetry of the crystal structure is 622, the indices could be assigned with the opposite polarity, i.e. $(1, 0, \bar{1})$, and $(2, \bar{1}, 2)$. The indices are in accord with the models in the manuscript.

The statement on page 2, line 57 reads now: “The best way to assign the morphology is $\{11\bar{1}\}$ for isohedral faces on the bottom pyramid and enantiomorphous faces either $\{122\}$ or $\{212\}$, on the top.” We added on page 4, line 116: “The crystal face assignments have been validated by single-crystal X-ray diffraction (**Fig. S4**).”

2. In the introduction, the authors suggest that enantiomorphous MOF crystals can help us understand the link between symmetry and crystal habit. However, MOFs with such enantiomorphous morphologies are rare, as demonstrated in this work, while a few examples of inorganic crystals exhibit similar features (Science 2021, 372, 729–733). Do MOF crystals offer additional advantages, such as size, unique optical properties, or ease of morphology tuning?

Answer: Thank you for pointing out the beautiful study by Ben-Moshe et al. In revision, we stated on page 5, lines 125: “Chiral crystal shapes can form due to chiral molecules, crystal structures, and additives.”^{19,20} Ben-Moshe et al.

found that screw dislocation-mediated growth of tellurium chiral nanocrystals leads to chiral polyhedral shapes.²¹” As shown in our study, the MOFs can be functionalized with chromophores resulting in dichroic crystals (Page 5, line 144 and **Fig. 2I,J**). The chirality of the morphology can be altered as a function of the solvent composition (Page 5, line 127 and **Fig 2E-G, Fig. S6**). These two findings are clearly presented in the text. There are many studies related to the advantages of MOFs for various applications. We prefer not to go into a discussion about additional advantages of MOFs instead of other crystals. It is a fundamental study regarding the correlation of chirality of crystallographic structures and their morphologies.

References added on pages 15 and 16:

19. Orme, C. A., Noy, A., Wierzbicki, A., McBride, M. T., Grantham, M., Teng, H. H., Dove, P. M., DeYoreo, J. J., Formation of chiral morphologies through selective binding of amino acids to calcite surface steps. *Nature* 411, 775–779 (2001).
20. Lahav, M., Leiserowitz, L., The Story Behind the link between molecular chirality and crystal shape. *Helv. Chim. Acta* 106, e202200172 (2023).
21. Ben-Moshe, A., da Silva, A., Müller, A., Abu-Odeh, A., Harrison, P., Waelder, J., Niroui, F., Ophus, C., Minor, A. M., Asta, M., Theis, W., Ercius, P., Alivisatos, A. P., The chain of chirality transfer in tellurium nanocrystals. *Science* 372, 729–733 (2021).

3. In previous reports, the authors synthesized MOFs formed by 3d metal ions and the non-chiral AdDB ligand with different morphologies (Angew. Chem. Int. Ed. 2022, 61, e202205238). Mn-AdDB and Cd-AdDB (QW-MOF) crystallize in the chiral P622 space group, but Mn-AdDB exhibits symmetric hexagonal morphology. A discussion on the differences in coordination geometries among various metal centers and the role of metal ions in forming enantiomorphous morphologies is necessary.

Answer: The series of our reported single crystals exhibit a wide range of (seemingly multidomain) morphologies, including brainy crystals, double-decker flowers, and hollow crystals (as stated on pages 6, line 156). Refs: 9, 12, 15, 16. Most of these morphologies are not chiral, except for the double-decker flower morphology reported in ref 12: *Nat. Commun.* **11**, 380 (2020). It

would be too speculative to discuss the role of metal cations in forming enantiomorphous morphologies. A thorough discussion on the differences in coordination geometries among various metal centers and the role of metal ions in forming enantiomorphous morphologies would be more suitable for a review article that we are drafting. Mn-AdDB, as reported in *Angew. Chem. Int. Ed.* 2022, 61, e202205238, and QW-MOF are synthesized under different experimental conditions. Crystallization parameters can significantly affect morphology and dimensions. As we have shown in this manuscript and other studies, there are many parameters that influence crystal morphology. In this work, the solvent composition governs the chirality of the morphology – and not the metal cations.

4. In many examples of spontaneous chiral resolution, the Flack parameter is typically used to differentiate between enantiomorphous crystals. The Flack parameter of QW-MOF is 0.28, which could be due to small crystal size or poor diffraction quality, but it is also possible that some racemic twinning is present. Could chiroptical imaging with a complete polarimetric microscope be used to determine the purity of the crystal's chirality? Additionally, can the twisted bipyramidal crystals possibly arise from racemic twinning?

Answer: We agree with the reviewer that the Flack parameter may suggest either poor diffraction quality or the presence of racemic twinning. Over 100 crystals from different batches were collected for single crystal X-ray analysis at 100K (Page 8, line 179). We added the following text to page 10, line 240: “The chiroptical imaging technique provides spatially resolved information about optical activity across individual crystals. This technique allows us to distinguish between homochiral crystals and those potentially containing mixed enantiomorphs or domains arising from twinning.²⁷ Our imaging data reveal consistent oscillating sawtooth circular birefringence profiles across the entire crystal, which excludes racemic twins.’ Worthy of note, we do observe interpenetrated twinned crystals as shown by X-ray micro-computed tomography (microCT) (Page S5, **Fig. S5**).

5. Authors demonstrate the correlation between methanol proportion and crystal offset angle, and mention that solvent polarity might be one of the reasons. However, it lacks further experimental and theoretical support.

Answer: The experimental data clearly demonstrate that methanol content influences the chirality of the crystal morphology. This effect is observable through both scanning electron microscopy and optical imaging, and we have included these data in the manuscript to provide an intriguing example of how chirality can be controlled in this system. We appreciate the reviewer's comment regarding the possible explanation for the correlation between methanol proportion and the crystal offset angle. The addition of methanol undoubtedly alters the polarity of the medium, which could be a contributing factor. The ^{113}Cd NMR spectra of $\text{Cd}(\text{OAc})_2$ showed an upfield shift, suggesting CH_3OH solvation with concurrent reduction of coordinated counter-anions. This effect could have altered the reactivity of Cd^{2+} , resulting in different morphologies.

Fig. S7. ^{113}Cd NMR of $\text{Cd}(\text{OAc})_2$ in (a) DMF, (b) DMF (5% CH_3OH), (c) DMF (10% CH_3OH).

However, fully understanding this effect would require an extensive separate study involving many additional experiments, density functional theory (DFT) calculations, and molecular dynamics simulations to analyze solvent-crystal interactions. For example, differential solvent adsorption on specific crystal facets could play a role in the observed offset angle variation. Additionally, the starting metal salts may interact differently with various solvents, affecting their reactivity and potentially influencing the crystallization process. Investigating these effects in detail is non-trivial and beyond the scope of this already substantial body of work. In our revision, we have added PXRD data to the Supplementary Information confirming that the crystal

structure remains unchanged (Page S7, **Fig. S8**), while only the chirality of the morphology is affected by variations in solvent content. We believe these additional data will be of interest to readers and help clarify the impact of solvent composition on the system. We state on page 5, Line 133: “The ^{113}Cd NMR spectra of $\text{Cd}(\text{OAc})_2$ indicated CH_3OH solvation with a reduction in coordinated counter-anions, possibly altering Cd^{2+} reactivity and crystal offset angle (**Fig. S7**). However, PXRD measurements confirm that the crystallographic structure remains unchanged (**Fig. S8**). Therefore, the achiral solvent composition only impacts from the expression of the chiral crystal structure in the morphology.”

Fig. S8 Powder X-ray diffraction (PXRD) data of **QW-MOF**. Non-twisted (top) and twisted-crystals (bottom). Both samples were prepared at room temperature using layering. The non-twisted sample was obtained by using 10% methanol. The data was collected by measuring the samples in a rotating capillary along with the reaction solution. The data indicate a $P622$ space group.

6. The oscillating sawtooth profile associated with CR displays a positive slope for R-twisted crystals or a negative slope for L-twisted crystals. What about the results for symmetric bipyrimidine crystals or crystals with an offset angle close to 0? Is the difference between L- and R-twisted crystals attributed to the morphological chirality, or does it originate from the topological chirality (the P or M helical structure in L- or R-twisted crystals)?

Answer: Please note that the experiments demonstrating the correlation between methanol proportion and crystal offset angle were conducted using solvent diffusion rather than solvothermal reaction conditions. These crystals are relatively small and nearly impossible to manipulate for CD measurements. As seen in **Fig. 4,5** (Page 9 and 11), **Fig. S9** (Page S9) the oscillating sawtooth profiles reflect the optical responses from the top of the crystals toward the center. The oscillating sawtooth profiles remain unaffected by the specific area analyzed. The manuscript includes a study of the optical activity of ethylenediammonium sulfate (EDS) model crystals (Page 11, line 245, Page S7, S10, **Fig. S10, S11**). We demonstrate that our chiroptical imaging setup, equipped with a complete polarimetric microscope, can effectively distinguish between dextrorotatory and levorotatory enantiomorphs. The oscillating sawtooth profiles clearly originate from the chiral structure and not from the morphology. For further reference, please see Ref. 27: Tan, S. C. M., *The Chiroptics of Imperfect Crystals*, Doctoral Dissertation, New York University (2020).

Other:

1. On page 2, line 36, “is remains” should be “remains”.
2. On page 3, line 73, "3-left" should be corrected to 3-right.
2. In Table S2, the molecular formula should also include acetate counterions, and the decimal points for the R1 and wR2 values need to be corrected.

Answer: The typos on page 2 and 3 have been corrected. We have corrected the X-ray analysis and updated table S2 (Page S11).

Reviewer #2 (Remarks to the Author):

In this work, the authors demonstrate a direct correlation between enantiomorphous MOF crystals (named QW-MOF) and chirality at the molecular level. The crystal habits' geometry is linked to single crystal optical activity along low symmetry directions. The X-ray scattering from small QW-MOF crystals have been confirmed a hexagonal, enantiomorphous space group. In addition, the dissymmetric crystal forms of QW-MOF structures can be customized by solvent composition. Interestingly, the optical circular birefringence of the MOF enantiomorphs, observed through chiroptical imaging with a polarimetric microscope, has been used to correlate optical and morphological chirality. These results provide a new way to understand the molecular chirality of MOFs with their dissymmetric morphology. The reviewer encourages the acceptance of this manuscript after minor revisions.

1. The authors' method can only differentiate the correlation between crystals with distinct symmetrical morphologies. How does this differ from using polarizing microscopy to distinguish enantiomeric crystals?

Answer: This statement reflects a misunderstanding. The chiroptical properties are evident in the absence of the morphological dissymmetry. But the morphological dissymmetry permits a correlation between structure and properties. Typically, polarimetry distinguishes enantiomeric crystals after they have been dissolved (e.g. Pasteur and sodium ammonium tartrate). Or a crystal has plane parallel faces normal to an optical axis. Or a crystal is cubic (e.g. sodium chlorate). The technique here separates the chiroptical properties even in the presence of overwhelming linear anisotropy, the general presentation of most dissymmetric crystals.

2. The molecular-level helical chirality of QW-MOF materials needs confirmation through X-ray single crystal diffraction. How does this directly relate to its macroscopic morphology?

Answer: As stated on page 7, Line 178: “Over 100 crystals from different batches were collected for single-crystal X-ray analysis at 100 K.” and “The resolution is poor; the scattering drops off quickly at $2\theta > ca. 23^\circ$.” These crystals are inherently difficult to analyze by single-crystal X-ray diffraction.

The crystallographic model of QW-MOF is shown in Fig. 3 (Page 7), featuring chiral (left-handed) hexagonal channels. Optical images reveal a left (*L*) arrangement of the principal facets. Due to the poor X-ray resolution, we avoid directly correlating single-crystal X-ray diffraction data with macroscopic morphology. Instead, we use chiral optical imaging with a polarimetric microscope to correlate optical activity and morphological chirality. Our models of morphology are not connected to orientation matrix on diffractometer. As seen in **Fig. 4,5** (Page 9 and 11), **Fig. S9** (Page S9) the oscillating sawtooth profiles reflect the optical responses from the top of the crystals toward the center. The manuscript also includes a study on the optical activity of ethylenediammonium sulfate (EDS) model crystals (Page 11, line 245, Page S7, S10, **Fig. S10, S11**). Our chiroptical imaging setup - equipped with a full polarimetric microscope - effectively distinguishes between dextrorotatory and levorotatory enantiomorphs. The observed oscillating sawtooth profiles clearly originate from the chiral structure itself. For further reference, please see Ref. 27: Tan, S. C. M., *The Chiroptics of Imperfect Crystals*, Doctoral Dissertation, New York University (2020).

3.The MOF crystals in this study are highly perfect, and their morphology's dissymmetry correlates well with molecular chirality. Are the techniques and methods developed in this work also applicable to systems where crystals do not show obvious dissymmetry in their morphology?

Answer: Yes, the manuscript includes a study of the optical activity of ethylenediammonium sulfate (EDS) model crystals (Page 11, line 245, Page S7, S10, **Fig. S10, S11**). We demonstrate that the chiroptical imaging setup, equipped with a complete polarimetric microscope, can effectively distinguish between dextrorotatory and levorotatory enantiomorphs. For further reference, please see Ref. 27: Tan, S. C. M., *The Chiroptics of Imperfect Crystals*, Doctoral Dissertation, New York University (2020).

4.In Table S2, the value of 'Final R [$I > 2\sigma(I)$]' and 'R (all data)' are written incorrectly., e.g., 'R1 = 0. 0.2950, wR2 = 0. 0.5856', 'R1= 0.0.3109'.

Answer: Thank you. These typos have been corrected.